# Correlation between Thickness and Optical Properties in Nanocrystalline γ-Monoclinic WO₃ Thin Films

Renee J. Sáenz-Hernández [1,2,*], Guillermo M. Herrera-Pérez [3,*], Jesús S. Uribe-Chavira [1], María C. Grijalva-Castillo [3], José Trinidad Elizalde-Galindo [2] and José A. Matutes-Aquino [1]

1 Materials Physics Department, Centro de Investigación en Materiales Avanzados, Miguel de Cervantes 120, Chihuahua 31136, Chih., Mexico
2 Instituto de Ingeniería y Tecnología, Universidad Autónoma de Ciudad Juárez, Av. Del Charro 450 Norte, Ciudad Juárez 32310, Chih., Mexico
3 CONACYT—Centro de Investigación en Materiales Avanzados, S.C., Miguel de Cervantes 120, Chihuahua 31136, Chih., Mexico
* Correspondence: joselin.saenz@cimav.edu.mx (R.J.S.-H.); guillermo.herrera@cimav.edu.mx (G.M.H.-P.); Tel.: +52-6144394889 (R.J.S.-H.); +52-6144394890 (G.M.H.-P.)

**Abstract:** Results from the analysis of the variation of structural defects, such as oxygen vacancies indicate that by adjusting the thickness of the WO₃ films, fabricated by DC reactive sputtering, it is possible to modulate the oxygen vacancies concentration. This has a tremendous influence on the applications of these semiconductor materials. The thicknesses analyzed here are 42, 66, and 131 nm. After the annealing process at 500 °C, films were directly transformed to a stable γ-monoclinic crystal structure with $P2_1/n$ space group, with a preferential orientation in the (200) plane. Atomic force microscopy exhibits nanometer range particle size with the highest roughness and higher surface area for the thinner film. FTIR analysis shows the presence of characteristic bands of the double bond stretching vibrational modes (W=O) and stretching vibrations of the γ(W-O-W) bonds corresponding to the monoclinic WO₃. Raman bands located at 345, and 435 cm$^{-1}$ are ascribed to the presence of $W^{5+}$ species that induces the formation of oxygen vacancies VO. The thinner film shows a decrease in the optical indirect band gap attributed to the formation of oxygen vacancies in combination with $W^{5+}$ species that induce the formation of energy states within the forbidden band gap range.

**Keywords:** thin films; thicknesses; oxygen vacancies; structural defects; tungsten trioxide; optical properties; microstructural properties; semiconductor; DC reactive sputtering





## 1. Introduction

Semiconducting tungsten trioxide is a widely investigated material due to its good properties for a wide range of applications, from energy to medicine [1–5]. In recent years, much progress has been made in the manufacturing processes of WO₃, whether in the form of 1D, 2D, and 3D structures, having increasingly precise control over the composition, the crystalline structure, or the morphology. However, challenges continue to be presented to improve the understanding of the mechanisms that occur in the material to generate specific properties, and develop devices with advantages superior to those that exist until now [6].

Preparation techniques such as thermal evaporation [7], sol–gel [8], and laser ablation [9] have been used to deposit WO₃ thin films; compared to the other preparation methods, the sputtering method has the flexibility to change and control the deposition conditions (power, time, temperature, pressure, argon, and oxygen flows, and target-substrate distance), making it a convenient method to deposit films under different parameters. By changing these process variables, it is possible to control the microstructural characteristics for use in specific technological applications [10]. It is well known that the properties of thin films are highly dependent on their microstructure, composition, crystal defects,

and interfaces, and all these take place at the initial stages of film growth [11,12]. Apart from this, the impact of annealing also has a great influence on the physical properties of semiconducting oxides [13,14] that is why there is still a need to better understand the basic aspects of phase transformations of $WO_3$-based materials, particularly in thin films and nanostructures, because some of their applications require annealing and/or higher temperatures for their operation [15].

Due to the fact that $WO_3$ is a polymorphic material, it presents several transformations depending on the thermal treatment [16–18]. Previous reports [19–22] suggest that the $WO_3$ can be viewed as a defect perovskite-like $ReO_3$ structure with tilted and distorted corner-sharing $WO_6$ octahedra. Each W ion is octahedrally surrounded by six oxygen ions, and each oxygen ion is linearly flanked by two W ions [22]. This tilting and distortion of $WO_6$ octahedra leads to several structures, including monoclinic I ($\gamma$-$WO_3$), monoclinic II ($\varepsilon$-$WO_3$), triclinic ($\delta$-$WO_3$), orthorhombic ($\beta$-$WO_3$), tetragonal ($\alpha$-$WO_3$), and cubic $WO_3$ [23]. The most stable structure at room temperature is $\gamma$-monoclinic phase with the following lattice parameters: a = 7.297 Å, b1 = 7.539 Å, c1 = 7.688 Å, and $\beta$ = 90.91°. W-O bonds form zigzag chains in three directions with W-O-W angles of 158° and O-W-O angles of 166°. In the x direction, the bonds are of equal length, while in the y and z directions, they are alternately long and short [24].

Therefore, it is important to characterize and obtain detailed information on the structural and physical characteristics of the surface/interface of monoclinic $WO_3$ films as a function of preparation conditions. Hence, the motivation of the current work is to prepare $WO_3$ thin films prepared by the DC reactive cathodic evaporation technique and post-annealed at 500 °C for 1 min; to evaluate the effect of thickness on the structure (crystal phase, symmetry, and preferred orientation), microstructure (size, shape of nanoparticles, and roughness), and optical properties (band gap energy values). This work also considers structural defects such as oxygen vacancies, which are tuned by the change in the thickness of the $WO_3$ thin films. $WO_3$ has the unique property of being able to tolerate an abundance of oxygen vacancies in its lattice while maintaining a stable crystal phase, which has led to various oxygen-deficient forms [25–27]. To the best of our knowledge, few reports have shown significant interest in tuning the thickness and structural defect concentrations in monoclinic $WO_3$ thin films by adjusting controllable variables during the preparation of thin films such as annealing temperature.

## 2. Materials and Methods

### 2.1. Preparation of Substrates and WO₃ Thin Films

$WO_3$ thin films were grown on Corning 7059 glass substrates. Before deposition, glass substrates were ultrasonically washed and cleaned, first in isopropanol and then in acetone for several minutes to remove the possible contaminants present on them. A metallic target of tungsten trioxide (99.99% purity) of two inches in diameter was employed for the fabrication of the films. Using an ATC Orion 3 system from AJA International, the plasma was obtained with argon gas and prior to the film deposition, with the shutter closed, it was kept for 5 min to remove the last contaminants and impurities present on the uppermost layer of the target. Subsequently, for reactive deposition, pure oxygen gas was introduced to the deposition chamber, and for the reactive atmosphere, the argon and oxygen flow rates were adjusted in a ratio of $Ar/O_2$ 3:1, leaving a total chamber pressure of 0.66 Pa. All the thin films were deposited at room temperature, at a distance between the target and the substrate of 28 cm, with substrate rotation at 40 rotations per minute to maintain a good homogeneity of the films. A DC power supply was used in all depositions with an output of 60 W. The only different parameter during deposition was the time because it was modified to obtain different thicknesses of the thin films. These times were 5, 15, and 30 min. Once the thin films were obtained, they were characterized by X-ray diffraction, scanning electron microscopy, atomic force, and Raman spectroscopy; however, the as-grown films were amorphous. Then the annealing method was reproduced as described by Liu et al. [28], the $WO_3$ thin films were heated from room temperature up to 500 °C in air

for 1 min, at a heating ramp of 2 °C/min, and a cooling ramp of 2 °C/min down to room temperature in a Barnstead Thermolyne Type 47,900 furnace. Subsequently, once again, the same characterizations along with some others were made.

### 2.2. Characterization of $WO_3$ Thin Films

The structure and preferred orientation of $WO_3$ thin films were monitored by X-ray diffraction in a X'pert Pro PANalytical diffractometer (Marveln Panalytical, Cambridge, UK) with CuKα radiation (λ = 1.5418 Å, T = 293 K) using the grazing beam technique (GIXRD). All the samples were measured in a range from 20° to 60° with an angular step of 0.02°. The diffraction peaks were indexed using Crystal Impact Match!—Phase Analysis Software, Version 3.x [29]. Film thicknesses were determined using a JEOL JSM-7401F (JEOL, Akishima, Tokyo, Japan) field emission scanning electron microscope in cross-section mode and elemental analysis was realized by energy dispersive X-ray spectroscopy (EDX) (Octane Elect EDAX System) (EDAX, Warrendale, PA, USA). The films were sputter coated with Au for 20 sec prior to observation in the microscope. Surface characterization was carried out with an Oxford Instruments MFP-3D Infinity Atomic Force Microscope (Oxford Instruments, San Diego, CA, USA) operating in tapping mode to determine the film roughness with the help of the WSXM 5.0 software [30]. Infrared spectroscopy was performed using a Shimadzu IR Affinity 1S spectrometer (Shimadzu Corporation, Kyoto, Japan). The reflectance spectrums were obtained from 4000 to 450 $cm^{-1}$ using a Smiths Quest model ATR accessory. $WO_3$ Raman bands were identified using micro Raman scattering spectra from 100 to 900 $cm^{-1}$ using a Horiba spectrometer model LabRam HR Vis 633 with an He-Ne laser (wavelength 632.8 nm), with a resolution of 1 $cm^{-1}$. Lorentzian functions were considered using the Fityk program to obtain information about the peak center for each Raman band [31]. The optical properties such as band gap were obtained using transmission UV–visible spectra from 200 to 2500 nm. The measurements were performed with a dual-beam spectrophotometer Cary 5000 from Agilent Technologies.

### 3. Results

In this study, the influence of thickness variation of sputtered tungsten trioxide on the microstructural evolution and optical properties is analyzed.

### 3.1. X-ray Diffraction

After the annealing at 500 °C, the $WO_3$ thin films became polycrystalline as observed in the X-ray diffraction patterns obtained at a grazing incidence angle (Figure 1). The diffraction peaks were matched considering the monoclinic structure with P21/n (14) space group supported with the JCPDS 43-1035 [32]. No peaks related to impurities were detected. When the $WO_3$ thin films crystallize in the monoclinic $P2_1/n$ space group, $W^{6+}$ is bonded to six $O^{2-}$ atoms to form corner-sharing $WO_6$ octahedra. This structure has lattice parameters of a = 7.297 Å, b = 7.539 Å, c = 7.688 Å, and β = 90.91°. This monoclinic phase has been reported in nanocrystalline $WO_3$ materials at temperatures above 350 °C by different authors [28,33]. It can be noticed that all films show a preferential orientation in the (200) plane located at 2θ ~ 24.45°. According to the literature, this heat treatment induces a preferred orientation of the (200) plane indicating that it is the most stable thermodynamically [34]. Simchi et al. [35] suggested that the preferential orientation in the (200) plane implies coordination between the planes of $WO_3$ with maximum atomic density, that is, the tungsten atoms lie exactly within these planes that are parallel to the substrate. For Kwong et al. [34], the annealing temperature plays an important role in obtaining a preferred orientation, suggesting also that the annealing time has a minimal impact. Another feature is observed in the XRD peaks, as the thickness of film increases, it can be noticed that the peaks became sharper and stronger. This feature could be associated with an enhancement in the crystallization process.

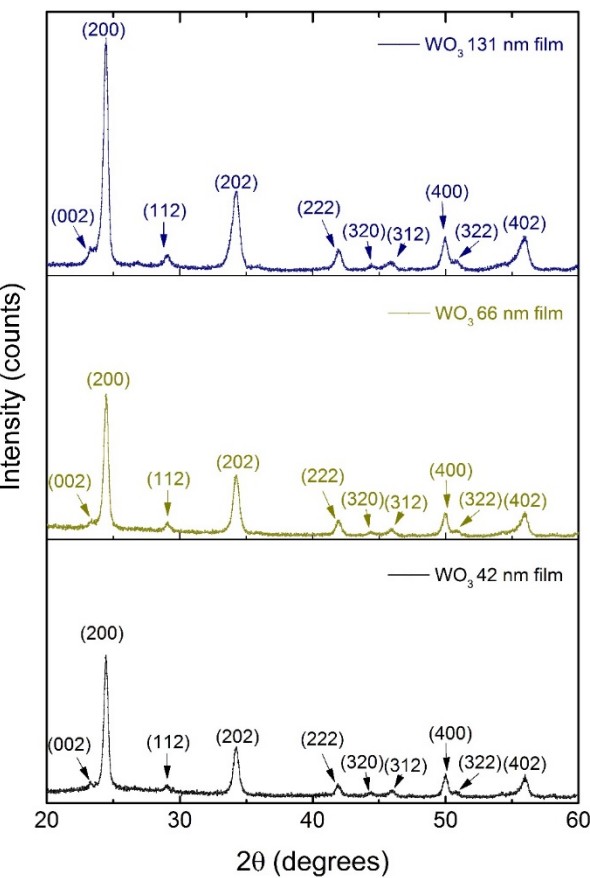

**Figure 1.** X-ray diffraction patterns of WO₃ thin films, with different thickness after annealing at 500 °C. All patterns were indexed to a monoclinic phase.

### 3.2. Scanning Electron Microscope (SEM)

The morphological characteristics of the $WO_3$ thin films, such as thickness and crack or pinhole presence were monitored using field emission scanning electron microscope in the cross-sectional mode. Figure 2a–c show the micrographs for monoclinic $WO_3$ thin films confirming the different thicknesses of 42, 66, and 131 nm when they were prepared as described above. In the analysis of the films by backscattered electrons, it was observed that all the films have a uniform and continuous film surface without cracks and pinholes. However, due to the resolution limit of the SEM images, other morphology characteristics such as the average particle size and shape distribution could not be identified with this technique. A more detailed structure should be analyzed with the help of AFM images. A compositional analysis was carried out on the surface film using energy dispersive X-ray spectroscopy (EDS). The Figure 2d, exhibits the characteristic W and O peaks present in the film. However, in the X-ray emission characteristic of W $M_\alpha$ overlaps with Si $K_\alpha$, which has a contribution due to the substrate used. In addition, it can be observed in the spectrum characteristic peaks of aluminum and barium contained in the substrate. A quantitative analysis of the $WO_3$ formation is also complicated by the fact that the O $K_\alpha$ peak also has a contribution from the substrate and the film [36]. The presence of oxygen and tungsten peaks is the evidence of formation of tungsten oxide on the surface and the growth of the films without impurities of other elements.

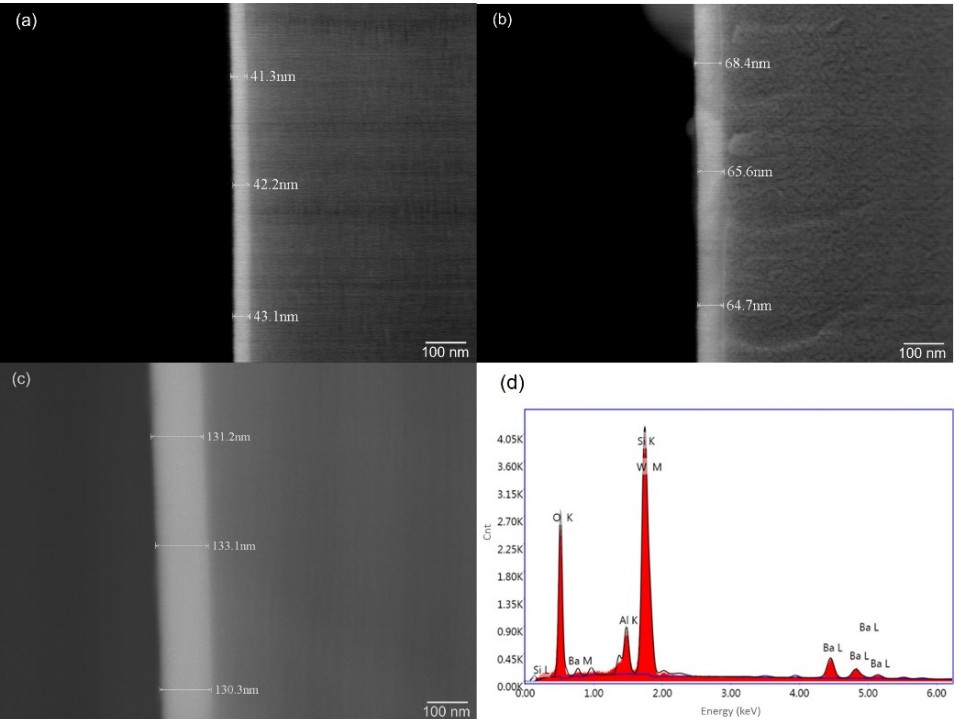

**Figure 2.** Cross-section SEM images of the monoclinic WO$_3$ thin films at different thickness (**a**) 42 nm, (**b**) 66 nm, (**c**) 131 nm, and (**d**) compositional analysis by EDS of the film.

### 3.3. Atomic Force Microscope (AFM)

The morphological evolution of monoclinic WO$_3$ thin films is depicted in the two- and three-dimensional AFM images (Figure 3). The typical morphology characteristic was elucidated, such as the average particle size, type-shape distribution, porosity, and surface roughness. The surface of the monoclinic thin films was smooth with small particles in the nanometer range that also exhibits uniform quasi-rounded type shape distribution for all the samples. Panel (a) is associated with 42 nm of thickness in the monoclinic film and it reveals smaller nanoparticles with a heterogeneous quasi-rounded granular particle shape distribution forming agglomerates of about 25 nm, giving rise to areas of porosity. The surface roughness for this sample is approximately 2.21 nm. The morphology surface for monoclinic WO$_3$ thin film with 66 nm of the thickness (panel b) shows a slight increase in the average particle size, in the range from 25 to 50 nm. This sample also exhibits a more homogeneous quasi-rounded particle shape distribution without the formation of agglomerates as in the thinner film. The surface morphology displayed in panel c can be observed as an increasing particle trend size as the thickness increases. Durante et al. [37] suggested that the increasing trend in particle size (based on Ostwald ripening [38]) is consistent for longer thin film deposition time. The AFM image also shows a dense surface with roughness decreasing with film thickness. The root-mean-square roughness (Rq) data for all samples were calculated using the WSxM 5.0 Develop 10.1 analysis software, in an area of 1.0 μm × 1.0 μm. Regarding the roughness variation as the thickness of films increases, the surface roughness decreases, from 2.21 nm to approximately 1 nm. According to the literature [39], the increase in thickness and decrease in the roughness of thin films both have an important influence on the morphology as well as the variation in the particle size and enhancement in the crystallinity.

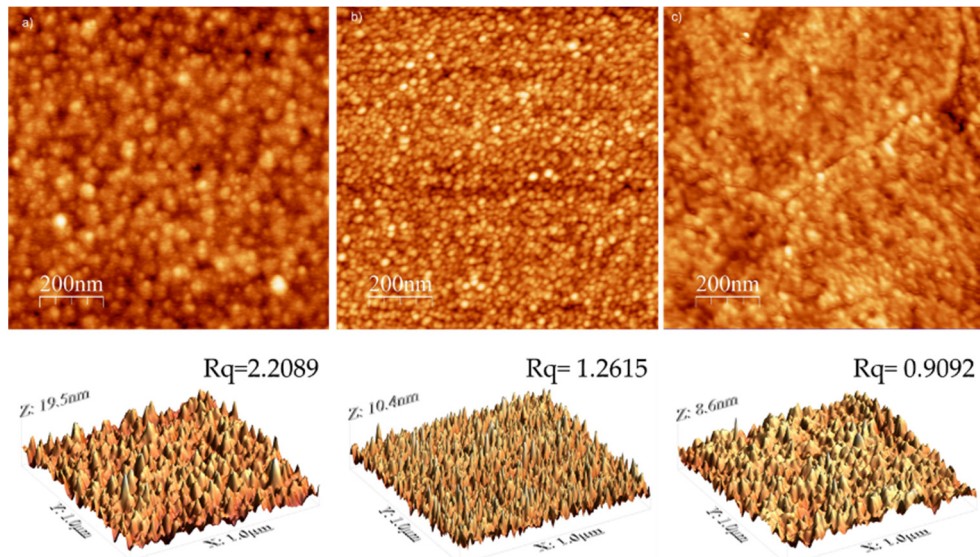

**Figure 3.** Two-dimensional and three-dimensional images from 1 μm × 1 μm surfaces obtained by AFM for monoclinic WO$_3$ thin films with thickness of (**a**) 42 nm film, (**b**) 66 nm film, and (**c**) 131 nm film, with their respective values of surface roughness Rq.

### 3.4. Infrared Spectroscopy (FT-IR)

Infrared spectroscopy was performed to identify the main bands of lattice vibrations. Figure 4 shows the FT-IR spectra in the range 450–4000 cm$^{-1}$ for the monoclinic WO$_3$ thin films which exhibits different thicknesses. One can observe that the spectra show the same main bands; however, there is a slight variation in transmittance percentage and a slight shift in wavenumber for the different thicknesses (see insert). Manciu et al. [40] suggested that this variation may be attributed to a gradual increase in crystallization of the films. The prominent bands are located at 671 cm$^{-1}$ (42 nm), 668 cm$^{-1}$ (66 nm), and 656 cm$^{-1}$ (131 nm) associated with γ(W-O-W) bridging modes of the WO$_6$ (octahedral) vibrations. The bands at 949 cm$^{-1}$ (131 nm), 947 cm$^{-1}$ (66 nm), and 941.5 cm$^{-1}$ (42 nm) correspond to the active stretching vibrations of the ν(W=O) bonds. Generally, the bands in the range 1637–3452 cm$^{-1}$ are attributed to the associated bending δ(HOH) and stretching ν(OH) vibrational modes of adsorbed water [41] on the oxygen vacancy sites [42].

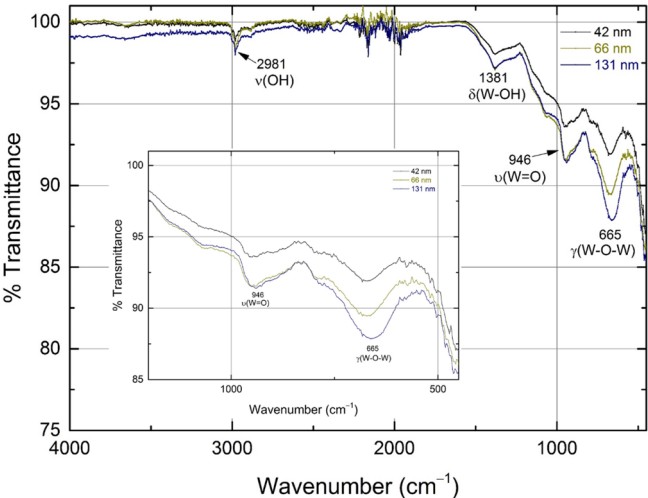

**Figure 4.** FTIR spectra for monoclinic WO$_3$ thin films with different thickness with an insert image in the range 450–1200 cm$^{-1}$.

### 3.5. Raman Spectroscopy

Raman spectroscopy, which provides identification of the crystalline phase present in $WO_3$, is also highly sensitive for monitoring structural changes such as the presence of lattice defects that contributes to the broadening of the peaks. Therefore, this characterization was carried out before the annealing treatment, without identifying any information on structural changes in the as-deposited films. Figure 5 shows the Raman results performed on the substrate without thin film and on the films without annealing treatment. After the annealing treatment, another spectral feature gives information about strains in the films. Figure 6a–c show the main Raman bands, after the annealing process, in the wavenumber range from 100 to 900 $cm^{-1}$, measured at room temperature, for the 42, 66, and 131 nm thin films. The Raman band indexation takes into account the irreducible representations reported previously in the literature [43]. The bands located at 270, 714, and 801 $cm^{-1}$ using the Fityk program are a reference to the existence of crystalline $WO_3$ with a monoclinic phase [41,44,45]. According to [33], the $P2_1/n$ symmetry belongs to the $C^5_{2h}$ point group. For this symmetry, the primitive unit cell contains 32 atoms so that there are 96 $\Gamma$-phonon modes [43]. The phonons can be classified according to the irreducible representations of this point group as $\Gamma$phonons = 24Ag + 24Au + 24Bg + 24Bu. Three of them are acoustic modes ($\Gamma$acoustic = Au+ 2Bu), 48 are Raman active modes (Ag and Bg), and 45 are infrared (IR) active modes (Au and Bu) [44]. Below 200 $cm^{-1}$, there are Raman modes attributed to lattice vibrational modes [46,47]. In fact, it can be observed that the sharpest peak at 130 $cm^{-1}$ is ascribed to the W-W mode that decreases in intensity when the film thickness increases. This is contrary to the $\delta$(O-W-O) band at 262 $cm^{-1}$, where the maximum intensity increases with increasing film thickness. In Figure 6 other sharp peaks located at 262 and 322 $cm^{-1}$ can also be noticed and are assigned to the bending vibration $\delta$(O-W-O) [47,48]. Li et al. [49] suggested that a broad peak centered at 262 $cm^{-1}$ indicates that the sample contains $W^{4+}$ species, and the Raman band located at 322 $cm^{-1}$ could be attributed to the $W^{5+}$-O single bond which indicates the presence of $W^{5+}$ defect states [50], which induces the presence of oxygen vacancies VO. Another Raman band could be observed in Figure 6 which is centered at 435 $cm^{-1}$ and is attributed to the $W^{5+}$=O bond [51]. For Kalanur et al. [27] this is attributed at the presence of oxygen vacancies accompanied by the presence of W mixed valence states ($W^{5+}$ and $W^{6+}$) introduced when the annealing in the air is performed and tend to modify the band gap as we will discuss in the UV–Vis characterization. Daniel et al. [52] suggested that in the range 750–950 $cm^{-1}$, the antisymmetric stretch of W-O-W bonds or symmetric stretch of (O-W-O) bonds are present. The intense peaks centered at 710 and 801 $cm^{-1}$ are typical Raman peaks that correspond to the stretching vibrations of the bridging oxygen [47,53,54]. They are assigned to W-O stretching ($\nu$), W-O bending ($\delta$) and O-W-O deformation ($\gamma$) modes, respectively [52,54]. Other two features in the Raman spectra. The first one is the slight red-shift of the three main Raman bands reported by Daniel et al. [55] centered at 807, 715, and 273 $cm^{-1}$ toward lower wavenumbers as summarized in Table 1. According to the literature [56,57], this red-shift suggests a shortening of the W-O bond confirming the presence of oxygen vacancies formed during the annealing treatment. The second feature is that the Raman spectrum for the thin film with higher thickness confirms an increase in the crystallinity due to the exhibition of well-defined bands with sharp and intense Raman peaks.

**Table 1.** Comparison between Raman shift for monoclinic $WO_3$ N-TF labeled as film 1 (42 nm); film 2 (66 nm) and film 3 (131 nm) with Raman band values reported by Daniel et al. [55].

| Reference [55] (cm⁻¹) | Film 1 (cm⁻¹) | Film 2 (cm⁻¹) | Film 3 (cm⁻¹) |
|---|---|---|---|
| 807 | 802.3 | 803.4 | 804.6 |
| 715 | 713.6 | 709.4 | 708.8 |
| 273 | 266.7 | 271.7 | 266.7 |

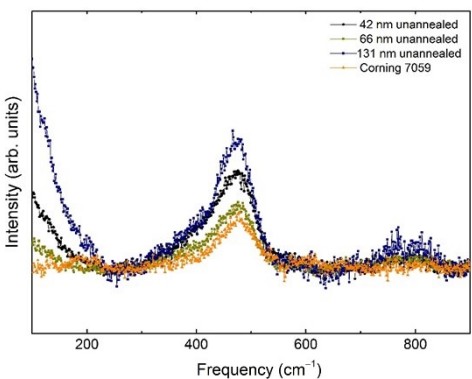

**Figure 5.** Raman spectra performed on the substrate 7059 without thin film and on the as-deposited films without annealing process.

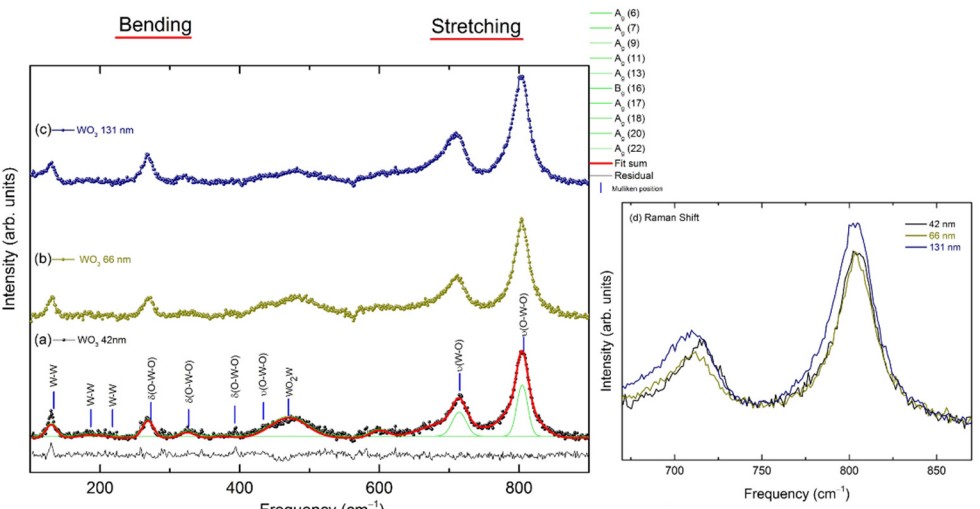

**Figure 6.** Raman spectra analysis using Fityk program for monoclinic WO$_3$ thin films with different thickness: (**a**) 42 nm, (**b**) 66 nm, (**c**) 131 nm, and (**d**) Raman shift.

### 3.6. UV–Vis Characterization

To gain more insight into the influence of structural defects such as oxygen vacancies in the optical band gap of monoclinic WO$_3$ thin films, UV–Vis characterization was performed. It is well known that bulk monoclinic WO$_3$ is an n-type semiconductor with a widely tunable band gap, ranging from Eg ~ 2.6–3.0 eV at room temperature [57]. The tunable band gap could be also associated with the introduction of W$^{5+}$ species accompanied by the formation of oxygen vacancies, which both act as shallow electron donors for n-type WO$_3$ [58] in the band gap. To monitor the variation of optical band gap energy for monoclinic WO$_3$ N-TF, the Tauc method was determined using the UV–Vis characterization through the following equation:

$$(\alpha\, h\nu)^{1/\gamma} = B(h\nu - Eg) \tag{1}$$

where h is Planck's constant, $\nu$ is the incident light frequency, B is a constant, $\alpha$ is the absorption coefficient, and Eg is the band gap energy. The $\gamma$ factor depends on the nature of the electron transition and is equal to $\frac{1}{2}$ or 2 for the direct and indirect transition band gaps. From the $(\alpha h\nu)^{1/2}$ versus photo-energy (h$\nu$) curves depicted in Figure 7, one can observe that an indirect band gap behavior is exhibited. Figure 6 also shows the evolution of $(\alpha h\nu)^{1/2}$ versus photo-energy (h$\nu$) curves of the absorbed light for the monoclinic WO$_3$ thin films with different thicknesses. It can be noticed in these curves that the energy band gaps were determined at the interception of the linear prolongation with the energy

axis [59]. The optical indirect band gap was 2.5 eV for the 42 nm, 3.1 eV for the 66 nm, and 2.9 eV for the 131 nm films.

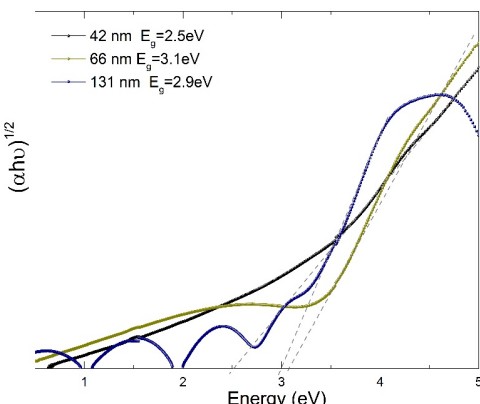

**Figure 7.** Optical indirect band gap (Eg) determination using the Tauc method for the WO$_3$ N-TF, which exhibits different thickness.

The smaller optical indirect band gap obtained for the thinner 42 nm film could be attributed to two main aspects: (1) the presence of structural defects such as oxygen vacancies [60] and (2) the influence of crystallite size. Recent reports [61] suggest that the oxygen vacancies, in combination with W$^{5+}$ species, induce the formation of energy states within the forbidden band gap range. This condition results in the narrowing of the band gap [62]. In fact, these new energy levels are localized below the conduction band minimum of bulk WO$_3$. The presence of defective energy states could act as electron "traps" to inhibit the electron transfer. The trapped electrons in the W$^{5+}$ states polarize the WO$_3$ lattice to induce polarons [63]. The moderate increase in oxygen vacancies in WO$_3$ causes a gradual decrease in the optical band gap [60]. In addition, film thickness is another crucial feature that determines the width of the optical band gap [64].

## 4. Discussion

During the thin film fabrication stage, there are several parameters that can be controlled, as mentioned above, and that makes reactive sputtering one of the most useful techniques for precision, reproducibility, and subsequently, scalability. Therefore, the method choice for material fabrication, substrate type, annealing process, and the film thicknesses, has a great influence on the microstructural properties in thin films and consequently on the optical properties of the material. According to the results obtained with the fabrication technique used, under certain conditions, it induces a morphology in granular-type thin films, since by this technique, it is possible to obtain different morphologies and porous microstructure [65–67]. It is clear that after the annealing process at elevated temperatures the crystalline structure is affected in some way, and in this case, the preferential orientation takes part by temperature changes, the process crystallinity influences differently by having different thicknesses. As indicated by Yousif and Khudadad [68], it could be attributed to the atoms moving along the substrate surface to touch the weak energy nucleation sites and growing preferentially in that direction. The presence of oxygen vacancies accompanied by the presence of W mixed valence states, monitored in the structural changes of this work, and by varying the thickness, represents a major strategy for the development of highly efficient WO$_3$-based materials, for control of the distribution or concentration of vacancies without involving complex methods [69].

## 5. Conclusions

Based on the experimental evidence presented in the current work, the following conclusion can be drawn. Using the reactive sputtering technique, monoclinic WO$_3$ thin films were prepared successfully with different thicknesses for different deposition times.

They were amorphous after deposition and crystallized after annealing at 500 °C. Then, the processing conditions to obtain the thin films such as the deposition process as well as the annealing treatment had an important influence on the microstructural and optical properties. All XRD patterns suggested the presence of monoclinic crystal structure with preferred orientation along the (200) direction. Cross-sectional SEM analysis confirmed the thickness of the films (42, 66, and 131 nm). Even though the heat treatment was the same for the different thicknesses, the temperature influenced it differently in each case, giving rise to smaller grain sizes at lower thicknesses. AFM images revealed surface roughness decreases as the thickness of films increased and also as the particle size increased. Raman band spectra indexation was performed considering a monoclinic phase with $P2_1/n$ space group. The localized Raman bands centered at 325 and 435 cm$^{-1}$ are associated with the presence of structural defects such as oxygen vacancies. The UV–Vis analysis suggested that the sample with less thickness (42 nm) presents a reduction in the optical indirect band gap. This reduction could be attributed to the presence of structural defects such as oxygen vacancies accompanied by $W^{5+}$ species.

**Author Contributions:** R.J.S.-H.; investigation, methodology, writing—original draft preparation, G.M.H.-P.; revised it critically for important intellectual content and approved the version to be published, J.S.U.-C.; formal analysis, M.C.G.-C.; validation, J.T.E.-G.; visualization, J.A.M.-A.; conceptualization and revised it critically for important intellectual content. All authors have read and agreed to the published version of the manuscript.

**Funding:** This research received no external funding.

**Institutional Review Board Statement:** Not applicable.

**Informed Consent Statement:** Not applicable.

**Data Availability Statement:** Not applicable.

**Acknowledgments:** The authors thank C. R. Santillán, A. Gonzalez, G. Rojas (XRD), O. Solis (AFM, Nanotech), C. Leyva (SEM, Nanotech), P. Pizá (Raman, Nanotech CIMAV), M. Orozco, and L. de la Torre (FTIR), for assistance in the characterization.

**Conflicts of Interest:** The authors declare no conflict of interest.

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
