# Peer review of "Correlation between Thickness and Optical Properties in Nanocrystalline γ-Monoclinic WO3 Thin Films"

_coatings, doi:10.3390/coatings12111727_

Round 1

Reviewer 1 Report

The paper presents a detailed study of WO3 thin films grown by DC reactive sputtering technique at different film thicknesses on the microstructuctural and optical properties of the films.

Raw 173: The authors use “nanoparticle” word when they are talking about the morphology of the films. Is this correct?

Raw 189: Please define the degree of crystallinity. Can be the degree of crystallinity calculated?

Author ResponseNovember 2nd, 2022

Ms. Alain Chen
Assistant Editor

Journal Coatings

Thank you for your letter on 26 October 2022, including valuable feedback for our manuscript titled “Effect of thickness on microstructural and optical properties of nanocrystalline -monoclinic WO3 thin films”. We appreciate the feedback from the referees, which was positive and provided helpful suggestions for improvement. We have addressed the referee’s comments and included a point by-point list of our responses and we have generated for each of them a report with the modifications. Please note that comments by the referee are labeled as (Q) and followed by the author’s response (R). The references can be found in the manuscript. The text that were added to the manuscript have been highlighted in yellow. Additionally, a grammar revision was carried out in the whole text.

Reviewer comments

Q1. Raw 173: The authors use “nanoparticle” word when they are talking about the morphology of the films. Is this correct?

R1:  In the literature it is referred that by means of the reactive sputtering technique it is possible to obtain different morphologies in the microstructure, we have referenced it in section 4, Discussion. Due to the results by AFM is that we appreciate morphology as nanoparticles, however, we have decided to change to the concept of granular type when we refer to the morphology of the microstructure in thin films.

Q2. Raw 189: Please define the degree of crystallinity. Can be the degree of crystallinity calculated?

R2: When we referred to the degree of crystallinity, we actually wanted to refer to the fact that during the annealing process, the crystallization process affects in different ways for each thickness, since as observed in the X-ray diffraction diffractograms, the intensity of the peaks varies as the thickness film increases, being thinner peaks and with greater intensity in the thicker film. We have modified for this case the concept to "crystallization process" during the writing in the hope of improving the wording.

Reviewer 2 Report

The paper with the title Effect of thickness on microstructural and optical properties of nanocrystalline -monoclinic WO3 thin films was not well-written. 

(1) The abstract was not well organized

(2) The introduction does not seem to show the research problem clearly

(3) Writing style of some of the citations was uncommon (e.g. line 41)

(4) The manuscript was not easy to understand.

(5) The results were not presented well

(6) The discussion need to improve

Author ResponseNovember 2nd, 2022

Ms. Alain Chen
Assistant Editor

Journal Coatings

Thank you for your letter on 26 October 2022, including valuable feedback for our manuscript titled “Effect of thickness on microstructural and optical properties of nanocrystalline -monoclinic WO3 thin films”. We appreciate the feedback from the referees, which was positive and provided helpful suggestions for improvement. We have addressed the referee’s comments and included a point by-point list of our responses and we have generated for each of them a report with the modifications. Please note that comments by the referee are labeled as (Q) and followed by the author’s response (R). The references can be found in the manuscript. The text that were added to the manuscript have been highlighted in yellow. Additionally, a grammar revision was carried out in the whole text.

Reviewer comments

The paper with the title Effect of thickness on microstructural and optical properties of nanocrystalline -monoclinic WO3 thin films was not well-written. 

R: We have changed the name to: "Correlation between thickness and optical properties in nanocrystalline g-monoclinic WO3 thin films", which we believe is more related to the topic of the article.

Q1. The abstract was not well organized.

R1: We have tried to improve the abstract, the changes in the new file are highlighted in yellow.

Q2. The introduction does not seem to show the research problem clearly.

R2: We have added information in the wording of the introduction, which is also highlighted in yellow.

Q3. Writing style of some of the citations was uncommon (e.g. line 41)

R3: We have reviewed the citations, we believe we are now complying with the format

Q4. The manuscript was not easy to understand.

R4: We have tried to improve the document; the changes in the new file are highlighted in yellow.

Q5.The results were not presented well

R5: Some modifications were made to the wording and images to try to improve the presentation of the results.

Q6.The discussion need to improve

R6: We write a short discussion section, which is displayed in the new file.

Reviewer 3 Report

"Effect of thickness on microstructural and optical properties of nanocrystalline gamma-monoclinic WO3 thin films"

The Authors present and discuss some results concerning tungsten trioxide (WO3) thin films grown by DC reactive sputtering technique, in particular the influence of specific parameters on the structure, morphology, and optical properties. The manuscript is poorly written, with several aspects that need careful and thorough revision.

* Title: meaningful
* Keywords: meaningful, yet insufficient (provide more)
* Abstract: not meaningful, the Authors need to further emphasize on the novelty and importance of their work;
use present tense (recommended) and avoid the use of acronym prior to providing the full name of the technique, mechanism, etc.

  1. Introduction

* this section is too short, missing important information; the Authors need to insist more on the novelty and importance of their approach with respect to literature and their previous work (some important publications are missing): further explain on your approach, and also consider providing more / further references to it.

* in the last paragraph of the section, the brief presentation of the work herein, needs to be presented with sufficient and relevant details, insisting on the novelty and importance of your approach; please rephrase this section accordingly.

  1. Experimental Methods

* as a general remark, this section is too short (incomplete) and the experimental methods need to be further discussed; however, there's no need to present well-known techniques (references will suffice, please provide more where available); avoid redundant text; provide further information on the compounds; use subsections (numbering).

  1. Results and discussion

* please start the section by describing the main aspects of your work - what do you seek and what is your plan in doing so (a few phrases will suffice).
* as a general overview / remark to this section: the Authors need to further discuss their results in a more correlated manner; provide more references to sustain your results (where available).
* please further present and discuss in text all figures and tables, provide references to sustain your results (where available).
* Figure 2. Cross-section SEM images  - provide adequate scalebars and details in the images (not clear).
* a final (last) paragraph of section 3 must be included, to provide the reader with a brief conclusion of your work / manuscript (and insist more on the novelty of your approach).

  1. Conclusion

* this section is poorly written and the Authors need to emphasize more on the novelty and importance of this approach, providing sufficient data relevant to this study; avoid bulletpoints (recommended).

Minor comments:
* please carefully proofread your manuscript for typographical errors, and for language, grammar and spelling.
* avoid the use of acronyms prior to providing the full name of the technique, mechanism, device, etc.

To conclude, the manuscript should be considered for publication only after careful revision.

Author Response

November 2nd, 2022

Ms. Alain Chen
Assistant Editor

Journal Coatings

Thank you for your letter on 26 October 2022, including valuable feedback for our manuscript titled “Effect of thickness on microstructural and optical properties of nanocrystalline -monoclinic WO3 thin films”. We appreciate the feedback from the referees, which was positive and provided helpful suggestions for improvement. We have addressed the referee’s comments and included a point by-point list of our responses and we have generated for each of them a report with the modifications. Please note that comments by the referee are labeled as (Q) and followed by the author’s response (R). The references can be found in the manuscript. The text that were added to the manuscript have been highlighted in yellow. Additionally, a grammar revision was carried out in the whole text.

Reviewer comments

Q:

The Authors present and discuss some results concerning tungsten trioxide (WO3) thin films grown by DC reactive sputtering technique, in particular the influence of specific parameters on the structure, morphology, and optical properties. The manuscript is poorly written, with several aspects that need careful and thorough revision.
* Title: meaningful
* Keywords: meaningful, yet insufficient (provide more)
* Abstract: not meaningful, the Authors need to further emphasize on the novelty and importance of their work; use present tense (recommended) and avoid the use of acronym prior to providing the full name of the technique, mechanism, etc.

1. Introduction

* This section is too short, missing important information; the Authors need to insist more on the novelty and importance of their approach with respect to literature and their previous work (some important publications are missing): further explain on your approach, and also consider providing more / further references to it.

* In the last paragraph of the section, the brief presentation of the work herein, needs to be presented with sufficient and relevant details, insisting on the novelty and importance of your approach; please rephrase this section accordingly.

2.Experimental Methods

* As a general remark, this section is too short (incomplete) and the experimental methods need to be further discussed; however, there's no need to present well-known techniques (references will suffice, please provide more where available); avoid redundant text; provide further information on the compounds; use subsections (numbering).

  1. Results and discussion

* please start the section by describing the main aspects of your work - what do you seek and what is your plan in doing so (a few phrases will suffice).
* as a general overview / remark to this section: the Authors need to further discuss their results in a more correlated manner; provide more references to sustain your results (where available).
* please further present and discuss in text all figures and tables, provide references to sustain your results (where available).
* Figure 2. Cross-section SEM images  - provide adequate scalebars and details in the images (not clear).
* a final (last) paragraph of section 3 must be included, to provide the reader with a brief conclusion of your work / manuscript (and insist more on the novelty of your approach).

  1. Conclusion

* this section is poorly written and the Authors need to emphasize more on the novelty and importance of this approach, providing sufficient data relevant to this study; avoid bulletpoints (recommended).

Minor comments:
* please carefully proofread your manuscript for typographical errors, and for language, grammar and spelling.
* avoid the use of acronyms prior to providing the full name of the technique, mechanism, device, etc.

To conclude, the manuscript should be considered for publication only after careful revision.

R: We have carried out each one of your accurate suggestions, so we are sending to you the new file, highlighting in yellow all the changes we have made and we hope we have made an improvement.

Thank you very much for your suggestions.

Reviewer 4 Report

The authors have studied the effect of thickness on crystal structure, microstructure and optical properties of WO3 films that are prepared by DC reactive sputtering technique and post-annealing treatment. They find the structural defects, such as oxygen vacancies, can be tuned by varying the thickness of WO3 film, which are closely related to the optical properties. The submission is worthy of publication after the following comments are considered. 

1. Whether the conclusion is affected by the preparation method of WO3. How about the WO3 film prepared by other methods ?

2. Is annealing treatment the only way to post-treat WO3 films? If not, the differences between high temperature treatment and other available methods should be discussed to make the presentation more convincing.

3. According to the author’s description, the film with thinner thickness shows smaller particles, but the formation of agglomerates gives rise to porous structure. What’s the possible reason behind this conflicting result? 

4. In order to observe the peaks of the IR spectrum more clearly, it would be better to optimize Fig. 4.

5. In order to observe the structural changes of the films before and after the annealing treatment more clearly, the Raman spectra of the WO3 films before the annealing treatment should be added in Fig. 5 for comparison.

Author Response

November 2nd, 2022

Ms. Alain Chen
Assistant Editor

Journal Coatings

Thank you for your letter on 26 October 2022, including valuable feedback for our manuscript titled “Effect of thickness on microstructural and optical properties of nanocrystalline -monoclinic WO3 thin films”. We appreciate the feedback from the referees, which was positive and provided helpful suggestions for improvement. We have addressed the referee’s comments and included a point by-point list of our responses and we have generated for each of them a report with the modifications. Please note that comments by the referee are labeled as (Q) and followed by the author’s response (R). The references can be found in the manuscript. The text that were added to the manuscript have been highlighted in yellow. Additionally, a grammar revision was carried out in the whole text.

Reviewer comments

Q1. Whether the conclusion is affected by the preparation method of WO3. How about the WO3 film prepared by other methods?

R1. We have tried to address this question with the variables that can be controlled during the process of thin film fabrication by the reactive sputtering technique (reactive cathodic evaporation). Reference was also made to literature dealing with this technique. For example, in the introduction section, we added the following paragraph to refer to your accurate observation:

Preparation techniques such as thermal evaporation [7], sol-gel [8] and laser ablation [9] have been used to deposit WO3 thin films; compared to the other preparation methods, the sputtering method has the flexibility to change and control the deposition conditions (power, time, temperature, pressure, argon and oxygen flows and target-substrate distance), making it a convenient method to deposit films under different parameters. Under these manipulation conditions in the process, it is possible to control the microstructural characteristics for use in specific technological applications [10]. It is well known that the properties of thin films are highly dependent on their microstructure, composition, crystal defects, interfaces, and all these take place at the initial stages of film growth [11, 12]. Apart from this, the impact of annealing also has a great influence on the physical properties of semiconducting oxides [13, 14] that is why there is still a need to better understand the basic aspects of phase transformations of WO3-based materials, particularly in thin films and nanostructures, because some of their applications require annealing and/or higher temperatures for their operation [15].

Q2. Is annealing treatment the only way to post-treat WO3 films? If not, the differences between high temperature treatment and other available methods should be discussed to make the presentation more convincing.

R2. We have tried to address this question in the sense that "post-treatment" refers to inducing the crystallinity process. In the case of our obtained films, all were made under the same conditions and did not form a defined crystalline structure, so we carried out the heat treatment to generate crystallization in this way.

Q3. According to the author’s description, the film with thinner thickness shows smaller particles, but the formation of agglomerates gives rise to porous structure. What’s the possible reason behind this conflicting result? 

R3. In the literature [65-68] it is referred that by means of the reactive sputtering technique it is possible to obtain different morphologies and the associated porosity in the microstructure.

Q4. In order to observe the peaks of the IR spectrum more clearly, it would be better to optimize Fig. 4.

R4. Figure 4 has been modified by adding an insert for the bands referred to in the text.

Q5. In order to observe the structural changes of the films before and after the annealing treatment more clearly, the Raman spectra of the WO3 films before the annealing treatment should be added in Fig. 5 for comparison.

R5. The Figure 5 was added to show the Raman results on the WO3 thin films before the annealing process and the substrate without film. The existing Raman figure was changed to number 6.

Figure 5. Raman spectra performed on the substrate 7059 without thin film and on the as-deposited films without annealing process.

Round 2

Reviewer 2 Report

I think the authors have revised the manuscript according to the previous comments and suggestions of the reviewers.  I would recommend accepting this manuscript. 

Reviewer 3 Report

"Effect of thickness on microstructural and optical properties of nanocrystalline -monoclinic WO3 thin films" - original title

"Correlation between thickness and optical properties in nano-2 crystalline -monoclinic WO3 thin films" - revision 1 title

The Authors have correctly addressed most of the issues raised during the peer-review procedure. The manuscript is now acceptable for publication in MDPI's journal "Coatings".